# Delay and Acceleration Threshold of Movement Perception in Patients Suffering from Vertigo or Dizziness

**DOI:** 10.3390/brainsci13040564

**Published:** 2023-03-27

**Authors:** Michel Toupet, Caroline Guigou, Cyrielle Chea, Maxime Guyon, Sylvie Heuschen, Alexis Bozorg Grayeli

**Affiliations:** 1Otolaryngology Department, Dijon University Hospital, 21000 Dijon, France; 2Centre d’Explorations Fonctionnelles Otoneurologiques, 75015 Paris, France; 3EA 7535, ImVia—Laboratory of Imagery and Artificial Vision, Burgundy University, 21000 Dijon, France

**Keywords:** movement perception, sound–movement delay, acceleration threshold, vertigo, dizziness, multisensory integration, persistent postural-perceptual dizziness

## Abstract

Background: The objective was to evaluate the delay and the acceleration threshold (AT) of movement perception in a population of patients suffering from dizziness and analyze the factors influencing these parameters. Methods: This prospective study included 256 adult subjects: 16 control and 240 patients (5 acute unilateral vestibular loss, 13 compensated unilateral loss, 32 Meniere diseases, 48 persistent postural-perceptual dizziness (PPPD), 95 benign paroxysmal positional vertigo (BPPV), 10 central cases, 19 bilateral vestibulopathy, 14 vestibular migraine, and 4 age-related dizziness). Patients were evaluated for the sound–movement synchronicity perception (maximum delay between the bed oscillation peak and a beep perceived as synchronous, PST) and AT during a pendular movement on a swinging bed. Results: We observed higher PST in women and in senior patients regardless of etiology. AT was higher in senior patients. AT was not influenced by etiology except in patients with bilateral vestibulopathy who had higher thresholds. AT was related to unipodal stance performance, past history of fall, and stop-walking-when-talking test. Conclusions: Delay and acceleration thresholds appear to be coherent with clinical findings and open insights on the exploration of symptoms that cannot be explained by routine otoneurological tests.

## 1. Introduction

The impact of dizziness in the general population is very high and increases with age [1]. Not only the loss of the vestibular function [2] but also other sensory deficits such as visual [3], proprioceptive [4], and auditory [5] disturbances seem to increase this risk. At a higher level, multisensory integration seems to influence the risk of falls. Indeed, in fall-prone senior individuals, a higher audiovisual stimulus onset asynchrony indicates possible disturbances in the processing of these inputs in comparison to healthy subjects of the same age [6].

In routine practice, exploring the vestibular function mainly consists of evaluating the reflex pathways in the brainstem and the cerebellum (e.g., the vestibulo ocular reflex, cervical and ocular vestibular-evoked myogenic potentials) or the overall balance strategy by dynamic posturography. Only the subjective visual vertical has a significant cortical implication among routine tests [7]. Hence, routine vestibular exploration underestimates the central participation in vertigo and does not reflect the clinical findings in several incapacitating diseases such as vestibular migraine [8] and persistent postural-perceptual dizziness [9]. It would be interesting to develop a new instrumental tool to assess central involvement in vertigo or postural disorders.

The delay and the threshold of movement perception involve both peripheral sensors and complex central integrators [10]. The acceleration threshold refers to the point at which the patient no longer perceives any movement (it corresponds to the downward threshold estimation method). The movement perception threshold is the time at which the patient indicates that the sound and the peak are synchronous.

Early works [11,12] focused on the movement perception threshold as an indicator of vestibular function. These studies did not investigate the role of the central nervous system in modulating the threshold and were soon replaced by more peripheral explorations such as the caloric test [13].

During the past decade, a better understanding of the pathophysiology of vertigo has allowed experts to focus on the central multisensory processing and to define new clinical entities such as vestibular migraine [8] and persistent postural-perceptual dizziness (PPPD) [9]. However, exploration tools are lacking in the clinical routine. In the light of several experimental reports [6,14], the idea of exploring movement perception appears interesting in addition to the peripheral tests. In a previous study, we showed that measuring sound–movement synchronicity and acceleration thresholds on a swinging rehabilitation bed yielded reproducible results in young healthy individuals [15].

We hypothesized that the perception of sound–movement synchronicity and the acceleration threshold during a pendular movement on a rehabilitation bed could be influenced by age, balance performances, and the etiology of the vertigo, especially those involving central vestibular processing. We also hypothesized that in case of bilateral vestibulopathy (BVP), patients would have higher uncertainty in the estimation of the movement delays and the acceleration thresholds if the visual and somatosensory inputs were suppressed.

The objective of this study was to evaluate the delay and the acceleration threshold of movement perception in a large population of vertiginous patients and analyze the factors influencing these parameters.

## 2. Materials and Methods

This prospective monocenter study included 256 subjects. The group included 16 healthy adults and 240 consecutive patients suffering from vertigo in a tertial referral center (Table 1). There was no age difference between men and women (57.9 ± 1.40 for women versus 57.1 ± 2.28 for men, not significant, unpaired *t*-test, n = 256).

All patients underwent a complete otoneurological workup including clinical history of dizziness and past falls, clinical examination, caloric and rotatory chair tests, visual fixation index test, saccades, gaze, and pursuit analysis, subjective visual vertical, audiometry, and cervical vestibular-evoked myogenic potentials, as well as a cranial MRI in selected cases. Diagnostic categories were based on this workup as follows: Age-related dizziness was defined by age > 75 years old and spontaneous dizziness, no evident deficit of canal or otolith function, and no identifiable neurologic abnormality.Bilateral vestibulopathy (BVP) was defined according to the Barany Society criteria: A horizontal angular vestibulo-ocular reflex (VOR) gain on both sides < 0.6 (angular velocity 150–300°/s) and/or the sum of the maximal peak velocities of the slow-phase caloric-induced nystagmus for stimulation with warm and cold water irrigations on each side < 6°/s and/or the horizontal angular VOR gain < 0.1 during sinusoidal stimulation on a rotatory chair (0.1 Hz, Vmax = 50°/s) and/or a phase lead > 68 degrees with a time constant of <5 s [16].Cured benign paroxysmal positional vertigo (BPPV) was defined according to von Breven et al. [17].Central disorders were defined as vertigo, dizziness, or unsteadiness associated to abnormal ocular pursuit control and/or gaze nystagmus and/or dysmetric saccades and/or absent ocular fixation and/or abnormalities of central vestibular pathways on MRI [18].Persistent postural-perceptual dizziness (PPPD) was defined by unsteadiness > 3 months, exacerbation by upright position, self- or visual-environment movements, significant functional handicap, and symptoms not better explained by any other disorder [9].Acute unilateral vestibular loss defined by a canal paresis on bicaloric test (>30% asymmetry of the sum of the 2 stimulations measured by the slow-phase velocity of the nystagmus on videonystagmography) and video head impulse test (vHIT, gain < 0.7 on at least one canal on the same side) for less than 30 days.Compensated unilateral vestibular loss was defined by a duration > 30 days, no rotatory vertigo and no spontaneous nystagmus.Probable Meniere’s disease was defined according to the Meniere’s disease diagnostic criteria [19].Vestibular migraine was defined according to Barany Society criteria [20].The control subgroup comprised healthy adult volunteers without any vestibular or auditory complaints or past medical history. This group did not undergo audiovestibular workup and was only tested on the swinging bed.

All patients provided their informed and written consent. The experimental protocol was approved by the local ethical research committee (CPP Est III) and the ANSM (number: 2015-A01053-46).

In addition, all subjects underwent the measurement of motion perception delay and acceleration threshold on a swinging bed.

### 2.1. Experimental Set-Up

A preliminary study was conducted to evaluate and validate the swinging bed [15]. Thirty healthy young adults without past medical history of balance disorders or hearing disabilities were tested on the swinging bed to evaluate the distribution of the above parameters and their repeatability (16 men and 14 women with a mean age of 32 years, range: 20–61). A second series of test–retest was carried out several days after the first (mean delay between series was 13 ± 2.1 days, range: 2–50). Four subjects were lost to follow-up for the second trial. Measures were conducted in the same manner as in the patients. During this study, the mean acceleration threshold was 9.2 ± 4.60 cm/s and the range width of the synchronous perception interval was 535 ± 190 ms. During the test–retest evaluation in the same trial, an acceptable reproducibility was found for the acceleration threshold and a good-to-excellent reproducibility was found for all parameters related to sound–movement latency.

The device was composed of a swinging bed suspended from a 2.5 m high gantry. Sound and friction were minimized by ball-bearings on the rotation axis. The radius of the oscillation was 2.4 m. Preliminary tests showed a 1% variation of this radius as a function of the weight of the subject. The swinging movement was initiated by a manual backward traction of the bed and a silent release. For the measurement of acceleration threshold perception, the amplitude of this initial displacement was controlled by a laser beam projected on a scale on the ground.

In order to measure the latency of the movement perception, an infrared detector was placed on the ground to detect the passage of the bed at its lowest point at each cycle. This device was connected to a processor and a loudspeaker, enabling the system to produce a beep (5 ms, 80 dB SPL) at the beginning of each oscillation (patient’s head at its highest position, peak). The delay between the peak and the beep could be adjusted by the operator by 50 ms increments.

The patient was installed in a supine position on the bed. The arms were placed along the body, and the legs were stretched. The nose pointed to the ceiling. Preliminary experiments excluded a possible effect of wind during the swinging movements. For the acceleration threshold (AT) measurements, the bed was pulled 8 cm backwards and released silently, letting it oscillate freely until a natural break. The patient was asked to indicate at which point he/she did not perceive any motion (descending threshold estimation method). At this point, the deviation from the equilibrium point was measured in centimeters. This deviation (d, cm) was converted to maximal tangential acceleration (a, cm/s^2^) by the following formula: $$a=9.81\times \left(\frac{d}{2.40}\right)\times 100.$$

To measure the movement perception delay, we evaluated the range of sound–peak delays in milliseconds, which produced a synchronous perception. The bed was pooled 10 cm backward from its equilibrium point and released. The delay between the beep and the peak was systematically varied from −750 to +750 ms in 50 ms increments. The patient was asked to indicate whether the sound and the peak were synchronous. A synchronous perception was noted for several delay values in all patients. For this study, we recorded the peak-sound threshold (PST) at the upper limit of this interval (Figure 1). The threshold was defined by an increment yielding a positive response followed by 2 negative responses to the following increments. Three successive test iterations separated by a 2 min break were conducted for this parameter. The average test duration was approximately 20 min.

In addition, patients underwent a timed unipodal-stance test, and the results were categorized as >5 s, <5 s, or impossible to hold [21]. A stop-walking-when-talking test was also administered according to Hyndman and Ashburn [22]. Briefly, patients were accompanied to the waiting room (25 m away) while questioned on their medication. The test was positive when the patients stopped walking during conversation while the examiner continued. Results were categorized as positive or negative.

While all patients underwent the swinging bed test, everybody did not complete the entire test battery. Consequently, the sum of n values in different categories may be less than 256, and they are indicated for each test and analysis.

### 2.2. Statistical Tests

Values were expressed as mean ± standard error of the mean (SEM). Data were analyzed by Graphpad prism (Graphpad Software Inc. V 5.01, La Jolla, CA, USA). A *p*-value < 0.05 was considered as significant. The n-values varied depending on the number of patients who completed each test, and some patients did not manage to complete all the tests. All subgroups were tested for normal distribution by Kolmogorov–Smirnov test.

Quantitative variables with multiple subgroups were analyzed by a mixed-effects model. In the case of repeated measures (iterations of the same test), a mixed-effect model for repeated measures (MMRM) was employed. In this case, we did not assume sphericity and a Greenhouse–Geisser correction was applied to the models. Unpaired comparisons of continuous variables were conducted by a Kruskal–Wallis test (to compare 3 or more groups) or a Mann–Whitney test (to compare 2 groups). For multiple comparisons, Tukey’s or Dunn’s tests were performed and adjusted *p*-values for multiple comparisons were provided.

The reliability of the PST measurements was evaluated by Cronbach’s alpha. A possible correlation between PST and acceleration threshold was tested by Pearson’s r.

The statistical tests and the reported parameters are detailed in Table 2.

## 3. Results

There was no difference between the parameters measured in the preliminary study and those in the control group concerning the PST (mean PST: 44 ± 35.9 ms, n = 16 for the control group versus 50 ± 90.2 ms, n = 30 for the preliminary study, Mann–Whitney test, *p* = 0.5409) and the AT (5.3 ± 1.12 cm/s^2^, n = 16 for the control group versus 9.2 ± 1.03 cm/s^2^, n = 30 in the preliminary study, Mann–Whitney test, *p* = 0.0765).

In the whole population, PST increased with iteration: 45.3 ± 12.16 ms for the first iteration (n = 238), 56.9 ± 11.59 ms for the second (n = 231), and 80.3 ± 13.99 for the third in the entire group (n = 229) (*p* = 0.0013, MMRM). We noted a good internal consistency between the three trials (global Cronbach’s alpha = 0.87, average R = 0.71). PST increased with iteration in both men and women, but appeared to be higher in women regardless of the etiology (Figure 2, repeated-measures, mixed-effects model, *p* = 0.0064 for the effect of iteration, *p* = 0.0153 for the effect of gender, and *p* = 0.867 for interaction).

Age also appeared to influence PST. This threshold was higher in senior patients than in younger individuals (Figure 3, MMRM, *p* = 0.0150 for the effect of age, *p* = 0.1156 for the effect of iteration, and *p* = 0.8561 for interaction).

We could not observe a relation between the PST and past history of falls (average PST = 67 ± 13.2 ms, n = 160 in nonfallers versus 60 ± 21.2, n = 67 in fallers, Mann–Whitney test, *p* = 0.6284) or between the PST and the stop-talking-when-walking test (58 ± 11.7, n = 210 in negative versus 88 ± 41.5, n = 28 in positive, Mann–Whitney test, *p* = 0.7751). In contrast, PST tended to be higher in groups with a lower unipodal-stance performance (47 ± 11.5, n = 199 for stance > 5 s, versus 130 ± 39.9, n = 32 for stance < 5 s, and 157 ± 90.0, n = 7 for impossible, Kruskall–Wallis test, *p* = 0.0670).

Diagnostic categories also appeared to influence the PST (Table 3). Interestingly, in PPPD, higher PST values were recorded in comparison to cured BBPV and to compensated unilateral loss (Table 3).

In patients with bilateral vestibular loss, seven patients (37%) could not provide a consistent response during the PST evaluation at the first trial, and this number increased with iterations (Table 3).

Acceleration thresholds were not affected by gender (6.9 ± 0.52 cm/s^2^, n = 159 in women versus 8.0 ± 0.90, n = 77 in men, *p* = 0.6021, Mann–Whitney test). However, they tended to be higher in senior patients (9.7 ± 9.60 cm/s^2^, n = 68 ≥ 70 years versus 6.4 ± 5.46, n = 152 for <70 years, *p* = 0.0812, Mann–Whitney test). As expected, patients with bilateral vestibulopathy had higher thresholds than those in the other diagnostic categories did (21.5 ± 14.91 cm/s^2^, n = 11 versus 5.3 ± 4.47 in controls, n = 16, adjusted *p* = 0.0033, Kruskall–Wallis test followed by Dunn’s test for multiple comparisons to control group), but other diagnostic categories such as unilateral deficit did not seem to modify this parameter (Figure 4).

Acceleration thresholds tended to be in accordance with the unipodal-stance test results (6.6 ± 5.87 cm/s^2^, n = 197 for >5 s, 10.1 ± 11.24, n = 30 for <5 s, and 9.8 ± 4.31, n = 8 for impossible, *p* = 0.0608, Kruskall–Wallis test), with the risk of fall (6.3 ± 5.61 cm/s^2^, n = 160 in nonfallers versus 8.6 ± 7.60 in fallers, n = 66, *p* = 0.0232, Mann–Whitney test), and with the stop-walking-when-talking test (10.2 ± 8.01 cm/s^2^, n = 26 in positive group versus 6.8 ± 6.59, n = 209 in negative group, *p* = 0.0229, Mann–Whitney test). It was noteworthy that the acceleration thresholds were not correlated to PST, indicating the independence of these two measures (Pearson r = 0.0562, 95% confidence interval: [−0.076, 0.187], R^2^ = 0.0031, *p* = 0.4066, n = 220, Pearson correlation test).

## 4. Discussion

In this study, the delay of movement perception as evaluated by PST and the acceleration threshold appeared as two different aspects of the central processing of balance since they were influenced by different parameters. Acceleration thresholds were mainly related to bilateral loss and fallers (based on the results of the unipodal-stance test, the risk of fall, and the stop-walking-when-talking test), while PST was influenced by gender and by diseases known to increase the sensitivity to motion, such as PPPD. We also observed that PST increased significantly with iteration in women and in younger subjects.

Synchronization of different sensory inputs (i.e., visual, somatosensory, auditory, and vestibular) is necessary in order to perceive in a coherent and realistic manner and to react appropriately to the environment [14]. To measure the delay between a movement and its perception, we decided to compare sound and movement since other options such as a motor or verbal response would have added an extra delay and variability to the response. Comparing sensory inputs other than sound to the movement perception was excluded since they also participate in the detection of body displacements.

The comparison between sound and vestibular input processing delays has been widely studied [14,23]. A movement perception has a longer processing delay compared to a sound perception: in fact, a sound has to be presented after an unpredictable movement in order to be perceived as simultaneous [23]. In pendular movements, with a predictable periodicity, anticipation appears to modify the delays [24]. Hence, in our protocol, sounds emitted before the movement reference point (head at its peak) were perceived as synchronous.

As we showed in the preliminary study on healthy subjects [15], the range of delays for which the sound and the oscillation peak are perceived as synchronous is around a negative figure (oscillation peak after the beep), suggesting a certain degree of anticipation. We showed that the upper border of this range (i.e., PST) provided consistent results in the control subjects in test–retest. Consequently, in this study, we explored only the PST, because exploring the entire interval from the peak-sound to the sound-peak thresholds would have been too long to perform in a clinical setting and on dizzy patients. We also showed that the width of the synchrony interval is potentially interesting. This might be a subject for further studies in patients.

The human brain employs past sensorineural experiences to anticipate and predict the future. Evidently, this anticipation is crucial in the calibration of movements and balance control [25]. In general, this type of prediction applies to the estimation of gravity-based movements [26], and rhythmic or oscillatory inputs [27]. While the vestibular system has a relatively short reaction time for postural and visual controls, the analysis of acceleration information is relatively slow since it is organized in a multilevel system (perception and then cognition), and needs the confrontation of several entries [10,14]. Estimating the perceived sound–movement delay in this study deals more with the cognition than with perception since it has to confront two different sensory modalities and anticipate the peak of a periodic oscillation, similar to the music perception [28]. The apparent variability in our measures during a pendular movement may be explained by the fact that the movement peak can be easily estimated (and anticipated) but the blunt peak of the movement adds imprecision to the estimation of the exact peak by the subject. The anticipation in periodic stimuli, which has been studied and modelized by other authors [23], explains smaller and even negative PST values. PST represents the upper limit of the delay interval, which is perceived as synchronous and represents the timing of vestibular input at the conscious level.

We noticed an increase in PST variation with iteration in patients with bilateral vestibulopathy. In accordance with other publications [28], this observation suggests that a disturbed perception of movements hampers the estimation of time intervals. The uncertainty generated by an altered sensory input has been widely studied; for a review, see [29]. Evaluating the uncertainty is based on a central modelization of the sensory input sequence and its timing. This model can be applied to enhance the prediction and the detection of the input in a bottom-up pathway. An altered input detection greatly hampers these evaluations in patients with BVP.

Higher PST values in PPPD are also suggestive of central processing alterations of the vestibular perception. Other publications have suggested that there are disturbances of multisensory processing in PPPD [30,31] and vestibular migraine [32,33,34]. In an fMRI study, Van Ombergen et al. showed modifications of the brain’s functional connectivity in the right temporal gyrus and the occipital lobe, which are implicated in visual and vestibular networks in PPPD patients [30]. These data were supported by voxel-based morphometry and demonstrated structural brain modifications in the occipital and the temporal cortex, but also in the hippocampus and the prefrontal regions implicated in multisensory integration [31].

The difference in PST between men and women and the apparent relation with age were also suggestive of a central processing involvement in this parameter. Several studies have already reported the effect of gender on line orientation, space orientation, and mental object rotation [35,36,37]. We have also previously shown that the visual attraction during the subjective visual vertical evaluation, equivalent to the visual dependency in the rod-and-frame test, was also higher in women and increased with age [38]. However, the influence of gender on these capacities remains unclear.

In contrast to the PST, the acceleration threshold appears to be processed more at a peripheral level since other sensory entries (sound and vision) are minimized. Contrarily to the PST, this threshold was higher in bilateral vestibulopathy and in those with a risk of fall, as evaluated by the stop-talking-when-walking test [22]. Additionally, the PST and the AT were not correlated, suggesting their independence. 

It is noteworthy that the range of the estimated acceleration thresholds was consistent with previous published data [39] and with the clinical findings. They were also relatively homogeneous in the control patients.

The potential relationships between age, gender, diagnostic categories, stance and walking performances, and swinging bed parameters make any mathematical modelization (e.g., multiple regression) hazardous. Like other vestibular tests, the swinging bed parameters should be considered in the context of each case and not as a predictor of a disease.

The measurement of movement perception has been reported in healthy subjects and in patients. In the 1940s, the duration of vertigo after a rotatory stimulation was measured and interpreted as an indicator of the vestibular function but not as a means by which to investigate the central processing [12]. More recently, linear [39,40] and parabolic [40] acceleration thresholds were measured in order to evaluate the otolithic function in healthy subjects and in patients with bilateral vestibulopathy. However, the experimental set-ups were complex and not applicable to the clinical routine. Our measures during the pendular movements of a swinging bed appear as a safe, relatively cheap, and quick method to investigate movement perception at the cortical level.

## 5. Conclusions

In conclusion, measuring the delay and the threshold of movement perception on a swinging bed provides interesting data on the cognitive aspects of vestibular processing, which are in line with already published data, and explores domains such as PPPD in which no routine explorations are available. This test is simple and noninvasive, making it applicable to dizzy patients and fallers.

## Figures and Tables

**Figure 1 brainsci-13-00564-f001:**
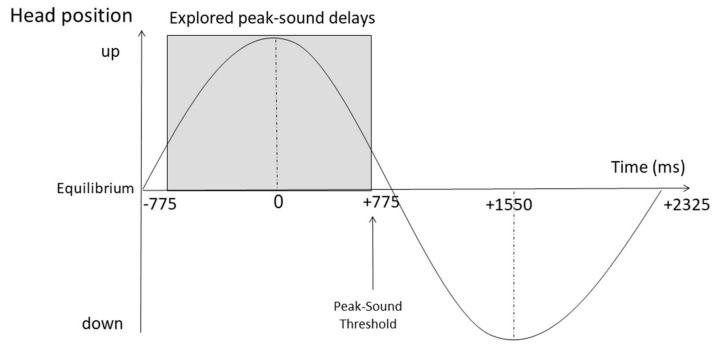
Relation between sound stimuli and bed oscillation. During bed oscillations a beep was generated by the electronic device with an adjustable time lag. The zero was defined as the peak of the oscillation (head at its maximal height). The time lag between the beep and the head peak was modified from −750 ms to +750 ms with 50 ms increments. Patients were asked to indicate whether the sound and the peak are synchronous. The explored interval is depicted in gray. The upper border was measured and defined as the peak-sound threshold.

**Figure 2 brainsci-13-00564-f002:**
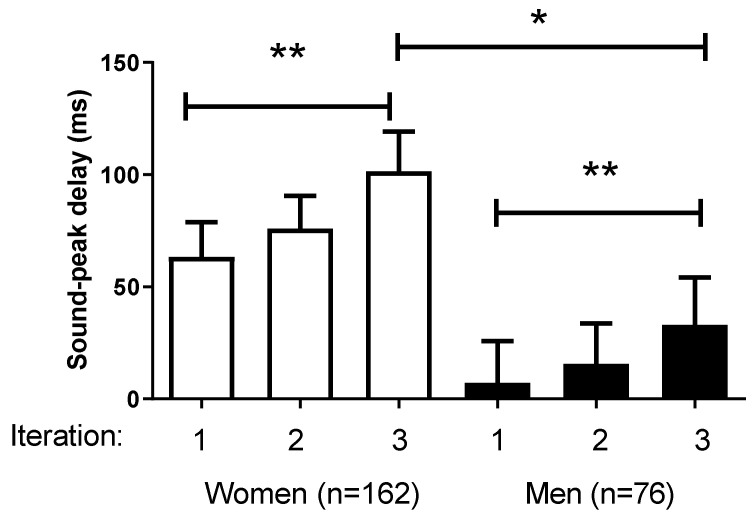
Effect of gender and iteration on PST. PST increased with iteration, and women had a higher PST than men did regardless of the etiology (repeated-measures and two-way mixed-effects model, ** *p* = 0.0064 for iteration and * *p* = 0.0153 for gender). PST: peak-sound threshold.

**Figure 3 brainsci-13-00564-f003:**
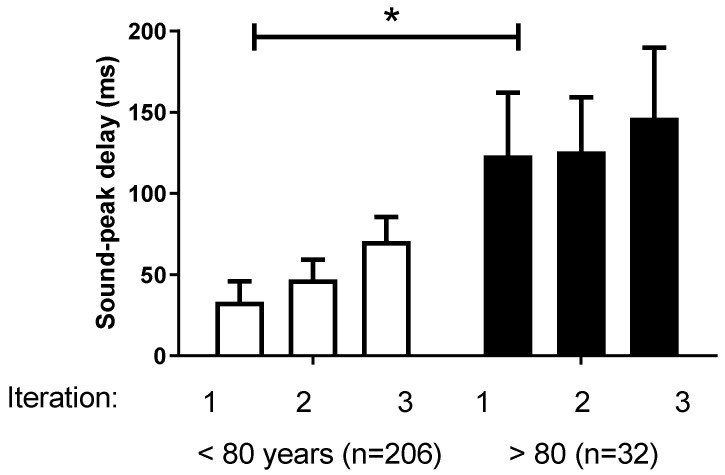
Effect of age and repetition on PST. Younger patients had a lower PST compared to older individuals (MMRM, * *p* = 0.0150 for the effect of age, *p* = 0.1156 for the effect of iteration, and *p* = 0.8561 for interaction). PST: peak-sound threshold.

**Figure 4 brainsci-13-00564-f004:**
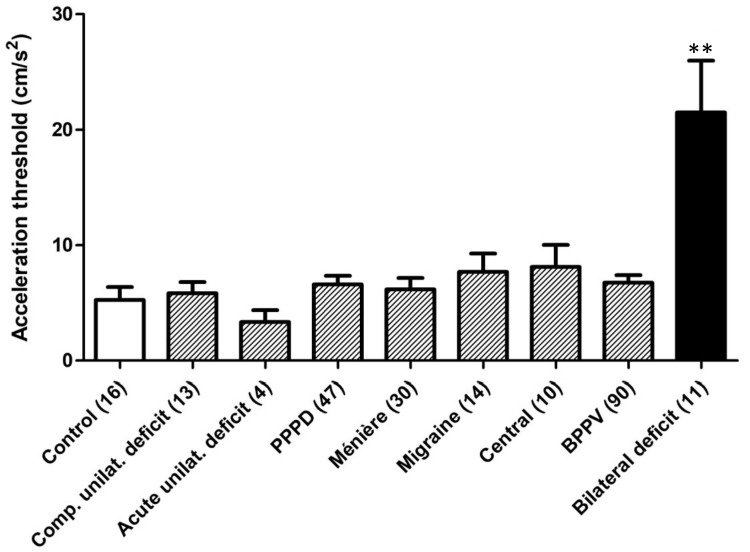
Acceleration thresholds as a function of etiology. Patients with bilateral vestibular deficit had higher acceleration thresholds (** adjusted *p* = 0.0033, Kruskall–Wallis test followed by Dunn’s test for multiple comparisons to control group). The threshold did not seem to be different from that of control subjects in other groups.

**Table 1 brainsci-13-00564-t001:** Population characteristics. SEM: standard error of mean.

Category	n	Age (Mean ± SEM)	Sex Ratio (Male/Female)
Age-related dizziness	4	89 ± 2.3	0.3
Bilateral vestibulopathy	19	66 ± 2.4	1.4
Cured Benign Paroxysmal Positional Vertigo	95	63 ± 1.7	0.3
Central Disorders	10	63 ± 5.1	0.4
Persistent Perceptual-Postural Dizziness	48	56 ± 2.9	0.5
Acute Unilateral Loss	5	54 ± 9.5	4
Compensated Unilateral loss	13	52 ± 5.8	0.9
Meniere’s Disease	32	52 ± 3.4	0.4
Vestibular Migraine	14	41 ± 4.0	0.4
Control	16	40 ± 5.1	0.8
Total	256	58 ± 1.2	0.5

**Table 2 brainsci-13-00564-t002:** Statistical tests performed in this study. BPPV: benign paroxysmal positional vertigo, UL: unilateral loss, PPPD: persistent postural-perceptual dizziness, BVP: bilateral vestibulopathy, VM: vestibular migraine, vs.: versus.

Parameters	Groups	Test
Average peak-sound threshold (PST, m/s) and acceleration thresholds (cm/s^2^)	Preliminary study controls (n = 30) vs. Current study controls (n = 16)	Mann–Whitney test
Iterative measures of PST	Iteration 1 (n = 238), iteration 2 (n = 231), iteration 3 (n = 229)	Mixed-effects model for repeated measures (MMRM) Global Cronbach’s alpha
Iterative measures of PST, effect of gender	Women (n = 162) vs. men (n = 76) Subgroups: iterations 1, 2, and 3	MMRM
Iterative measures of PST, effect of age	Subjects < 80 years (n = 206) vs. subjects >80 years (n = 32) Subgroups: iterations 1, 2, and 3	MMRM
Average PST, effect of falls	Nonfallers (n = 160) vs. fallers (n = 67)	Mann–Whitney test
Average PST, effect of stop-talking-when-walking (STWW) test	Negative (n = 210) vs. positive (n = 28) STWW test	Mann–Whitney test
Average PST, effect of timed unipodal-stance performance	Subgroups: stance > 5 s (n = 199) vs. stance < 5 s (n = 32) vs. impossible (n = 7)	Kruskal–Wallis test
Peak-sound threshold, effect of etiology	Etiology categories: Control (n = 16) Cured BPPV (n = 86) Central (n = 8) Acute UL (n = 5) Comp. UL (n = 13) BVP (n = 10) Meniere (n = 30) PPPD (n = 44) VM (n = 14) Age-Related (n = 3) Subgroups: iterations 1, 2, and 3	MMRM followed by Tukey’s posttest for multiple comparisons and *p*-value adjustment.
Acceleration threshold, effect of gender	Women (n = 159) vs. men (n = 77)	Mann–Whitney test
Acceleration threshold, effect of age	Subjects < 70 years (n = 68) vs. subjects > 70 years (n = 152)	Mann–Whitney test
Acceleration threshold, effect of etiology	Etiology categories: Control (n = 16) Cured BPPV (n = 90) Central (n = 10) Acute UL (n = 4) Comp. UL (n = 13) BVP (n = 11) Meniere (n = 30) PPPD (n = 47) VM (n = 14)	Kruskall–Wallis test followed by Dunn’s test for multiple comparisons to control group
Acceleration threshold, effect of falls	Nonfallers (n = 160) vs. fallers (n = 66)	Mann–Whitney test
Acceleration threshold, effect of STWW test	Negative (n = 209) vs. positive (n = 26) STWW test	Mann–Whitney test
Acceleration threshold, effect of timed unipodal-stance performance	Timed unipodal-stance performance, stance > 5 s (n = 197) vs. stance < 5 s (n = 30) vs. impossible (n = 8)	Kruskall–Wallis test
Acceleration threshold	Correlation to PST (n = 220)	Pearson correlation test

**Table 3 brainsci-13-00564-t003:** PST as a function of etiology categories. Values are presented as mean ± standard error of mean (n) in milliseconds. PST: peak-sound threshold. BPPV: benign paroxysmal positional vertigo, UL: unilateral loss, PPPD: persistent postural-perceptual dizziness, BVP: bilateral vestibulopathy, VM: vestibular migraine. Both etiology and iteration influenced PST (MMRM, *p* = 0.023 for etiology and *p* = 0.0085 for iteration, *p* = 0.3635 for interaction), * *p* = 0.0117 vs. cured BBPV, adjusted *p*-values for multiple comparisons Tukey’s test.

Etiology	Iteration 1	Iteration 2	Iteration 3	Average PST	Min	Max
Control	41 ± 54.2 (16)	53 ± 32.4 (16)	38 ± 42.0 (16)	44 ± 35.9 (16)	−217	283
Cured BPPV	33 ± 16.11 (90)	41 ± 18.3 (86)	58 ± 20.0 (86)	44 ± 16.1 (90)	−233	633
Central	−22 ± 42.6 (9)	44 ± 42.0 (9)	31 ± 51.7 (8)	19 ± 35.6 (9)	−167	167
Acute UL	130 ± 114.7 (5)	170 ± 75.2 (5)	190± 96.7 (5)	163 ± 88.3 (5)	−117	400
Comp. UL	−4 ± 41.0 (13)	31 ± 37.8 (13)	−4 ± 36.5 (13)	8 ± 34.8 (13)	−117	317
BVP	−71 ± 49.0 (12)	−123 ± 74.2 (11)	60 ± 143.5 (10)	−56 ± 72.8 (12)	−367	367
Meniere	5 ± 30.0 (30)	50 ± 25.3 (30)	62 ± 30.1 (30)	39 ± 26.1 (30)	−217	550
PPPD	121 ± 33.4 (45)	101 ± 28.4 (44)	142 ± 34.1 (44)	131 ± 31.3 (45) *	−300	750
VM	104 ± 61.7 (14)	139 ± 51.9 (14)	161 ± 60.9 (14)	135 ± 55.0 (14)	−67	550
Age-Related	138 ± 139.0 (4)	183 ± 136.4 (3)	250 ± 175.6 (3)	146 ± 94.6 (4)	−50	367
Total	45 ± 12.2 (238)	57 ± 11.6 (231)	80 ± 14.0 (229)	61 ± 11.4 (238)	−367	750

## Data Availability

The raw data supporting the conclusions of this manuscript will be made available by the authors, without undue reservation, to any qualified researcher.

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
