# Peer review of "Delay and Acceleration Threshold of Movement Perception in Patients Suffering from Vertigo or Dizziness"

_brainsci, 2023, doi:10.3390/brainsci13040564_

Round 1

Reviewer 1 Report

The authors investigated sound-movement synchronicity and acceleration threshold of movement perception measured on a swinging bed as two parameters representing cognitive aspects of vestibular processing on patients with vertigo or dizziness, using a commendable sample size involving 240 patients and 16 control. The design of the study seems to be rigorous. However, the analysis and presentation of the results are very confusing. Please see my detailed comments below:

11. The sample sizes in different analysis were keep changing throughout the manuscript. In the abstract, it said 240 patients and 16 control were involved. In Fig2, there were 229 women and 68 men, which did not sum up with 256. The total number in Fig3 was 238. The sample sizes were jumping in many other statements. For example,

“We could not observe a relation between PST, past history of falls (average PST = 67 ± 13.2 ms, n=160 in non-fallers versus 60 ± 21.2, n=67 in fallers, not significant, unpaired t-test) or stop-talking-when-walking test (58 ± 11.7, n=210 in negative versus 88 ± 41.5, n=28 in positive, not significant, unpaired t-test). In contrast, timed unipodal stance performance seemed related to PST (47 ± 11.5, n=199 for stance > 5 s, versus 130 ± 39.9, n=32 for stance < 5 s, and 157 ± 90.0, n=7 for impossible, p<0.05, one-way ANOVA).” 

Please clarify this.

2. In Fig.2, although the applied statistic was not significant of men group, to me the trend was the same as women group. Thus, I am not convinced by the statement saying PST significantly increased with iteration in women, but not in men.

3. It was misleading to use “trial” in Materials and Methods, then use ‘iteration” in Results.

4. “In this study, we found that the PST was 45.3 ± 12.16 ms for the first iteration, 56.9 ± 11.59 ms for the second and 80.3 ± 13.99 for the third in the entire group (n=229) and iteration did not seem to influence this parameter (one-way ANOVA).” Here the PST consistently increase from 45.3, to 56.9 and then 80.3. It is hard to believe iteration did not influence PST. Also, “in the entire group (n=229)” was also misleading. There should be 256 participants in the entire group.

5. “PST appeared to be higher in women than in men regardless of etiology (mean PST 80 ± 180.7, n=162 versus 22 ± 160.1, n=76, p<0.05, unpaired t-test).” The SEMs here were surprisingly large, which was not consistent with the error bar in Fig. 2. Similar issues in some other statements.

6. Fig. 2 suggested men had smaller PST, while Fig. 3 suggested younger participants had smaller PST. However, these tests were conducted separately, which could not rule out the cross influence between gender and age. For instance, could it because men involved in this study were in general younger than women?

7. Why use both negative and positive delays in measuring PST? Since average values were used in the comparison. Some patients may have big range around zero while some others may have small range around zero. Then the mean values could not differ them.

8. The N values in Fig. 4 were not consistent with Table 1.

9. Please give full name of abbreviations when they were first introduced, e.g., BPPV in abstract.

10. In table 1, maybe add (F/M) after the term of “sex ratio”.

Author Response

Thank you very much for your work

Reviewer 2 Report

The manuscript described a swing-based method which can be used to measure the synchronization between auditory sensation to movement perception, and the acceleration perception. The author researched parameters including gender, age, etiology of vertigo, and balance performance. The main concern for the study is that only one parameter was considered at one time. It is not clear if there are interactions among parameters. Is this appropriate to combine healthy group with all the other groups for the gender, age, and balance performance analysis? I suggest using a regression model and plug in all parameters to do a preview of the whole data set first.

Minor concerns:

1. Introduction: Would you mind defining “delay and threshold of movement”, “acceleration threshold” and “delay and the acceleration threshold of movement perception” in a clear way? The link between the conclusion about “routine vestibular tests underestimate the central participation in vertigo” and the objective of the study need more explanation.  

2. Table 1: Does Sex ratio mean male/female?

3. Results, Page 6, line 191. As mentioned, “There was no difference between the parameters measured in the preliminary study and those in the “normal group”. Would you mind providing data about this?

4. Results, Page 6, line 204. Is there any reason for using age 80 as a separator for “younger” and “senior” group?

5. Discussion, Page 8, line 264, the summary of the findings is not accurate. Why “Acceleration thresholds were mainly related to bilateral loss and fallers”? As the data shown, acceleration thresholds were also related to age, unipolar stance, and stop-walking-when-talking test. Same question goes with PST. What are the criteria for the “main” parameter?

Author Response

Thank you for your work

Round 2

Reviewer 1 Report

Thanks for addressing my comments. I have no further comments.

Author Response

Thank you very much for the quality of your revision.

Best regards

Reviewer 2 Report

Thank you very much for responding to my questions and suggestions.

Author Response

Thank you very much for the quality of your expertise.

Best regards.